# Role of Short- and Long-Lived Reactive Species on the Selectivity and Anti-Cancer Action of Plasma Treatment In Vitro

**DOI:** 10.3390/cancers13040615

**Published:** 2021-02-04

**Authors:** Kyriakos Sklias, João Santos Sousa, Pierre-Marie Girard

**Affiliations:** 1Université Paris-Saclay, CNRS, Laboratoire de Physique des Gaz et des Plasmas, 91405 Orsay, France; kyriakos.sklias@universite-paris-saclay.fr; 2Institut Curie, PSL Research University, CNRS, INSERM, UMR 3347, 91405 Orsay, France; 3Université Paris-Saclay, CNRS, UMR 3347, 91405 Orsay, France

**Keywords:** cold atmospheric-pressure plasma, plasma activated liquid, plasma cancer treatment, plasma medicine, direct treatment, indirect treatment, in vitro, RONS, short-lived species, long-lived species

## Abstract

**Simple Summary:**

One fundamental feature that has emerged from in vitro application of cold plasmas in cancer treatment is the key role of the liquid phase covering the cells. In the present work, we investigated the effect of direct and indirect plasma treatments on two cancer and three normal cell lines to assess the benefits of one treatment over the other in terms of death of tumor versus healthy cells. Our results demonstrate that indirect plasma treatment is as efficient at killing tumor cells as an appropriate combination of H_2_O_2_, NO_2_^−^ and acidic pH in ad hoc solutions, while sparing normal cells. However, direct plasma treatment is far more efficient at killing normal than tumor cells, and we provide evidence that short- and long-lived reactive species contribute synergistically to kill normal cells, while having an additive effect regarding tumor cell death. Collectively, our results call the use of plasma-activated liquid in cancer treatment into question.

**Abstract:**

(1) Plasma-activated liquids (PAL) have been extensively studied for their anti-cancer properties. Two treatment modalities can be applied to the cells, direct and indirect plasma treatments, which differ by the environment to which the cells are exposed. For direct plasma treatment, the cells covered by a liquid are present during the plasma treatment time (phase I, plasma ON) and the incubation time (phase II, plasma OFF), while for indirect plasma treatment, phase I is cell-free and cells are only exposed to PAL during phase II. The scope of this work was to study these two treatment modalities to bring new insights into the potential use of PAL for cancer treatment. (2) We used two models of head and neck cancer cells, CAL27 and FaDu, and three models of normal cells (1Br3, NHK, and RPE-hTERT). PBS was used as the liquid of interest, and the concentration of plasma-induced H_2_O_2_, NO_2_^−^ and NO_3_^−^, as well as pH change, were measured. Cells were exposed to direct plasma treatment, indirect plasma treatment or reconstituted buffer (PBS adjusted with plasma-induced concentrations of H_2_O_2_, NO_2_^−^, NO_3_^−^ and pH). Metabolic cell activity, cell viability, lipid peroxidation, intracellular ROS production and caspase 3/7 induction were quantified. (3) If we showed that direct plasma treatment is slightly more efficient than indirect plasma treatment and reconstituted buffer at inducing lipid peroxidation, intracellular increase of ROS and cancer cell death in tumor cells, our data also revealed that reconstituted buffer is equivalent to indirect plasma treatment. In contrast, normal cells are quite insensitive to these two last treatment modalities. However, they are extremely sensitive to direct plasma treatment. Indeed, we found that phase I and phase II act in synergy to trigger cell death in normal cells and are additive concerning tumor cell death. Our data also highlight the presence in plasma-treated PBS of yet unidentified short-lived reactive species that contribute to cell death. (4) In this study, we provide strong evidence that, in vitro, the concentration of RONS (H_2_O_2_, NO_2_^−^ and NO_3_^−^) in combination with the acidic pH are the main drivers of plasma-induced PBS toxicity in tumor cells but not in normal cells, which makes ad hoc reconstituted solutions powerful anti-tumor treatments. In marked contrast, direct plasma treatment is deleterious for normal cells in vitro and should be avoided. Based on our results, we discuss the limitations to the use of PAL for cancer treatments.

## 1. Introduction

To date, the three main anti-cancer therapies remain surgery, chemotherapy, and radiotherapy [1]. However, each of these therapies has its own advantages and drawbacks. For example, even if surgery is the most efficient to eradicate the tumor, it can be traumatic for the patient because of residual scars and/or ablation of an organ (e.g., breast during mastectomy). In contrast to surgery, chemotherapy and radiation therapy are only capable of killing a fraction of the tumor cells at each treatment. Therefore, more common use of radiation therapy is in combination with surgery and/or chemotherapy. Despite major improvements in the use of combined modality therapy, therapy resistance has been observed for every therapeutic regimen available today, including poly-chemotherapy, radiation therapy, immunotherapy, and molecular targeted therapy [2]. Deep RNA sequencing of the primary tumors has revealed the presence of resistant clones within the tumor before the beginning of therapy, suggesting that intra-tumor heterogeneity is an emerging mechanism of therapy resistance, not acquired under therapy selection, but inherent to tumor progression [2,3]. Consequently, there is still a need to develop new anti-cancer strategies to overcome those problems, and one of the most innovative aspects of cold atmospheric plasma (CAP) use over the last 15 years undoubtedly lies in cancer treatment.

Indeed, since the first proof-of-concept that CAP can damage mammalian cells in vitro [4], the effectiveness of CAP as an anti-cancer therapy, both in vitro and in vivo, has been reported in several human cancers [5]. This includes brain cancer, skin cancer, breast cancer, colorectal cancer, lung cancer, cervical cancer, leukemia, hepatoma, head and neck cancer, as well as osteosarcoma [5,6,7,8]. Moreover, CAP treatment can be combined with chemotherapy [9,10,11,12], nanoparticles [13,14,15,16,17], hyperthermia [18], radiotherapy [18,19], photodynamic therapy [20,21], and magnetic field [22]. Several clinical studies reported the use of CAP for treatment of (pre-) cancerous tissues [23] and two clinical trials using CAP are currently undergoing treatment of cervical intraepithelial neoplasia (ClinicalTrials.gov Identifier: NCT03218436) and Canady Helios cold plasma scalpel treatment at the surgical margin and macroscopic tumor sites (ClinicalTrials.gov Identifier: NCT04267575).

CAP is an ionized gas at near-room temperature, composed of a high number of reactive species, ions, electrons, metastable species, and also of the electromagnetic field, UV, and VIS radiation. Several CAP devices for medical applications have been described in the literature, and are based essentially on the use of atmospheric air (e.g., floating electrode DBD) or of noble gases such as helium (He) and argon (Ar) (e.g., DBD plasma jet), since these two gases result in the stable generation of glow-like discharges at low gas temperatures [24,25,26,27,28,29]. Admixture of molecular gases such as O_2_ is often used to enhance the generation of reactive radicals [29]. Since He or Ar plasma jets propagate in ambient air, molecular oxygen and nitrogen, and also water molecules, present in the air, diffuse inside the noble gas channel and interact with the excited states of the noble gas, producing various reactive oxygen (ROS) and nitrogen (RNS) species [30,31].

Among the various components produced in a plasma, ROS and RNS remain the most studied since these species have been shown to play a major role in the anti-cancer property of CAP [32,33,34]. In vitro, CAP treatment is usually performed in the presence of a liquid phase covering the biological target. Several types of short-lived (e.g., radical hydroxyl ^•^OH, anion superoxide O_2_^•−^, and singlet delta oxygen ^1^O_2_) and long-lived (hydrogen peroxide H_2_O_2_, nitrite NO_2_^−^, and nitrate NO_3_^−^) ROS/RNS have been detected in the gas and liquid phases [30,35,36,37,38]. Most of the reactive species found in the liquid phase are either primarily generated in the gas phase and at the gas-liquid interface or are the end-products of the transformation of primary ROS/RNS in liquid [30,31,38,39]. While H_2_O_2_ is a central player in the anti-cancer capacity of CAP treatment [40,41,42,43,44,45,46,47,48,49], we, and others, have shown that H_2_O_2_ and NO_2_^−^ act synergistically to induce cell death after plasma treatment [46,50,51].

The use of cold plasma in oncology is limited by the accessibility of the tumor. As such, we have to consider two types of plasma application modalities that are currently used to treat cancer cells: the direct and indirect treatments [52]. Direct treatment involves direct exposure of the biological target to plasma in the presence of a liquid (e.g., cancer cells in vitro) or not (e.g., superficial tumors in vivo), while indirect treatment involves the treatment of a liquid (e.g., saline solution like PBS, and cell culture medium), and subsequent application of these plasma-activated solutions onto the biological target, in vitro or in vivo [53,54]. In vivo, the treatment of superficial tumors such as skin tumors and head and neck tumors can be achieved by direct treatment. However, the big challenge for the plasma community is the treatment of non-superficial cancers. One approach is as an intra-operative adjuvant treatment [55]. Another method of delivering ‘‘plasma species’’ to deep tissues could be via plasma-activated liquids. The type of plasma application has several implications with regards to the nature of the physicochemical parameters that interact with the biological target. In the case of direct treatment, physical factors (ultraviolet, heat, and electromagnetic field) and chemical factors (long- and short-lived ROS/RNS) are present during the treatment, while only chemical factors, and among them essentially long-lived species such as H_2_O_2_ and NO_2_^−^, should be considered in indirect treatment.

While several groups have shown that, in vitro, both treatments were equivalent in inducing cancer cell death, in altering cell surface adhesion molecules or in inactivating enzymatic functions [46,56,57,58], others have reported that direct treatment is more effective than indirect treatment at killing tumor cells [59,60,61,62,63]. Moreover, there is still a debate regarding the selectivity of cold plasma at inducing cell death preferentially in tumor cells over healthy cells [46,49,64,65,66,67,68,69,70,71]. These discrepancies between reports can be attributed to the fact that the anti-cancer capacity of plasma-activated liquid depends on several factors such as the size of the wells in which cells are seeded, the volume of the treated liquid, the liquid composition (e.g., PBS versus culture medium), the gap between the plasma source and the liquid, the gas flow rate, the gas admixture, and the plasma device itself [48,71,72,73].

This study aims to investigate the effect of direct and indirect plasma treatments on two different head and neck cancer cell lines (CAL27 and FaDu), and three normal cell lines (primary human fibroblasts, primary gingival keratinocytes, and epithelial cells) in order to assess the benefit of using one treatment over the other in terms of tumor cell death versus sparing of healthy cells, and thus to bring new insights into the potential use of plasma-activated liquid in the field of cancer treatment.

## 2. Results

### 2.1. Influence of Gas Flow Rate and Treatment Distance on the Production of Long-Lived Species and pH Change in PAP

Since H_2_O_2_, NO_2_^−^ and NO_3_^−^ are the main long-lived reactive species created in plasma-activated PBS (PAP) [46,50], we first measured their concentration in PAP as a function of the treatment distance, treatment time, gas admixture and gas flow rate. Since the purpose of the present study is not to highlight the effects of all the tested parameters on the production of RONS in PBS (all these data will be presented in another manuscript, which is in preparation), we decided here to use the operating conditions that induce the highest and lowest concentrations in the PAP of each of the aforementioned species.

Our results showed that the concentration of these reactive species is significantly higher when the treatment distance is 8 mm (Figure 1a,b). Indeed, at a gas flow rate of 0.5 and 1 slm, the concentration of H_2_O_2_ + NO_2_^−^ + NO_3_^−^ was about 3.6× and 2.3× higher, respectively, for a treatment distance of 8 mm than 20 mm (Figure 1a,b). Among these three reactive species, H_2_O_2_ is the most sensitive to the treatment distance (but its concentration is relatively insensitive to the gas flow rate, at least for 0.5 and 1 slm). Indeed, H_2_O_2_ concentration dropped from ~1.6 mM at 8 mm of treatment distance to ~0.17 mM at 20 mm (Figure 1a,b). In marked contrast, NO_3_^−^ concentration did not change much for a treatment distance of 8 mm, ranging from 0.55 mM at 0.5 slm to 0.26 mM at 1 slm, while for 20 mm, the concentrations are almost identical (~0.35 mM). For a treatment distance of 20 mm, NO_2_^−^ concentrations are similar between 0.5 and 1 slm (0.51 mM versus 0.53 mM, respectively), while at 8 mm, a higher concentration of NO_2_^−^ was measured at 0.5 slm (~1.5 mM) than at 1 slm (~0.63 mM). Collectively, these data strongly support previous reports showing that changing the experimental conditions drastically impinges RONS production [48,71,72,73].

Some publications have reported that the plasma treatment of a liquid solution increases its pH [74,75], while we, and others, have reported that plasma causes acidification of the treated liquid [46,76]. We therefore addressed the effect of plasma treatment on the pH of PBS. We found a reduction of the pH of 0.55 and 0.4 units for a gas flow rate of 0.5 and 1 slm, respectively, at a treatment distance of 8 mm (Figure 1a, inset), and a reduction of 0.3 units for both gas flow rates of 0.5 and 1 slm at a treatment distance of 20 mm (Figure 1b, inset). The results demonstrated that plasma-treated PBS is slightly more acidic than untreated PBS.

### 2.2. Influence of Gas Flow Rate and Treatment Distance on Cancer Cell Death after Indirect Plasma Treatment

To investigate the cytotoxic effect of PAP as a function of the gas flow rate and the treatment distance, we decided to use two different head and neck cancer cell lines, CAL27 and FaDu, as this type of tumor has been already treated by plasma in humans [77] and are models of choice (easy to access, no invasive surgery, directly in contact with atmospheric air) for exploring the ability of CAP to reduce tumor growth and induce cancer cell death in vivo. The different cell treatment modalities used all along this manuscript are depicted in Figure 2.

At first, PBS solutions were exposed to CAP for 12 min at gas flow rates of 0.5 and 1 slm, treatment distances of 8 and 20 mm, and cancer cells were incubated for 1 h in these PAPs (indirect plasma treatment). Our results showed that the strongest toxicity is observed for a treatment distance of 8 mm. Indeed, cell viability ranged from 30% to 50% at this distance, while for a treatment distance of 20 mm, cell viability ranged from 50% to 70% (Figure 3). At a treatment distance of 8 mm, increasing the gas flow rate from 0.5 slm to 1 slm reduced the efficacy of indirect plasma treatments, while for a treatment distance of 20 mm, 1 slm is as efficient as 0.5 slm (Figure 3). These results strongly parallel the concentrations of RONS described in Figure 1, i.e., higher concentration of these species corresponds to lower cell viability. We also observed that CAL27 is slightly more resistant than FaDu to indirect plasma treatment.

### 2.3. Role of RONS on Cancer Cell Death

Since H_2_O_2_ and NO_2_^−^ act in synergy to induce cancer cell death [46,50], we investigated their inter-dependence in promoting cancer cell death. To do that, FaDu and CAL27 cells were exposed to two concentrations of H_2_O_2_ in PBS (0.8 mM and 1.6 mM), while the concentration of NO_2_^−^ was gradually increased up to 1.6 mM. At first, we observed that these two cell lines demonstrate a different sensitivity to H_2_O_2_ in standalone, with CAL27 being more resistant than FaDu (Figure 4). While NO_2_^−^, NO_3_^−^ or a combination of NO_2_^−^/NO_3_^−^ have virtually no toxic effect on the cell lines in the range of concentrations used in this study (cf. insets in Figure 4), increasing the concentration of NO_2_^−^ in the presence of H_2_O_2_ led to a concentration-dependent decrease of cell viability (Figure 4). The synergistic effect of these two reactive species on the viability of the cancer cells was calculated using Combenefit software (https://www.cruk.cam.ac.uk/research-groups/jodrell-group/combenefit) with Bliss, HAS, and Loewe models [78]. All models gave the same results (see Appendix A), strongly confirming that H_2_O_2_ and NO_2_^−^ act synergistically in inducing cancer cells death. As presented in Appendix A, this synergistic effect becomes more conspicuous for higher NO_2_^−^ concentrations. Collectively, the results demonstrate that optimum cancer cell death can be achieved at a well-defined concentration of H_2_O_2_ by adding the appropriate concentration of NO_2_^−^. For example, 60% cell viability for CAL27 is observed following incubation in H_2_O_2_ 1.6 mM or a combination of H_2_O_2_ 0.8 mM + NO_2_^−^ 0.2 mM (Figure 4a). Similarly, 30% cell viability for FaDu is observed following incubation in H_2_O_2_ 1.6 mM or a combination of H_2_O_2_ 0.8 mM + NO_2_^−^ 0.8 mM (Figure 4b).

Among the long-lived species present in PAP, H_2_O_2_ has been described as a master player in PAP-induced loss of cancer cell viability [41]. To further demonstrate that this is also true in our experimental conditions, cancer cells were incubated for 1 h in PAP containing or not catalase, which decomposes H_2_O_2_ into H_2_O and O_2_. Our results showed that the presence of catalase during PAP treatment completely prevents the toxicity of PAP (Figure 5a), and this fully correlates with a complete loss of H_2_O_2_ in PAP resulting from the addition of catalase (Figure 5b). These results underline the fact that H_2_O_2_ is an essential factor in PAP leading to cancer cell death.

### 2.4. Effect of Acidic pH in Combination with RONS on Cancer Cell Death

Having established that cancer cell death can be induced by a combination of H_2_O_2_ and NO_2_^−^, we then added another parameter to this cocktail of reactive species: the pH of PBS, whose value changes upon plasma treatment. For this study, we first considered a plasma treatment performed for 12 min at a gas flow rate of 0.5 slm and at a treatment distance of 8 mm, which are the conditions resulting in the highest production of RONS (Figure 1), the highest loss of viability for CAL27 and FaDu cell lines (Figure 3) and the strongest reduction in pH (inset of Figure 1). We therefore compared the cell viability of both cell lines after 1 h incubation in PBS pH 7.05 (untreated condition), 1 h incubation in PBS pH 6.5, 1 h incubation in PAP (plasma-treated PBS for 12 min at a distance of 8 mm and a gas flow rate at 0.5 slm), and 1 h incubation in a reconstituted buffer (PBS pH 7.05 or pH 6.5, containing 1.6 mM H_2_O_2_, 1.5 mM NO_2_^−^ and 0.55 mM NO_3_^−^, values derived from Figure 1a). We found that the acidic environment of PBS alone does not affect cell viability, while the reconstituted buffer (RB) at pH 7.05 has a strong effect, although not as strong as PAP (Figure 6). Finally, the combined effects of the acidic pH and the concentration of reactive species (RB at pH 6.5) recapitulate the effect of PAP on the viability of cancer cells (Figure 6).

We also performed the same analysis for a treatment distance of 20 mm, for which fewer RONS (Figure 1b), less loss of cell viability (Figure 3), and less pH reduction (inset of Figure 1b) were measured. Again, the reconstituted buffer (RONS in PBS plus adjusted pH) recapitulates the effect of PAP on the viability of cancer cells (see Appendix A). Together, these results demonstrate that, in addition to the concentration of RONS, the pH has to be considered to fully explain the toxicity of PAP during indirect plasma treatment. Most importantly, they also demonstrate that in vitro a reconstituted buffer is as efficient as indirect plasma treatment (PAP).

The concentration of the previously mentioned RONS and the pH value were measured as a function of the storage time of PAP. More specifically, the stability of H_2_O_2_, NO_2_^−^, NO_3_^−^, and pH in the treated solution was assessed 1, 3, 6, 12, 24 and 72 h after the plasma treatment. Concerning the concentrations measured immediately after the plasma treatment, no significant degradation (<10%) of these reactive species was observed for up to 3 days of storage at room temperature (23 °C) and the pH was also stable over time (see Appendix A). For the same time period and storage temperature, H_2_O_2_, NO_2_^−^ and NO_3_^−^, and pH were also stable in ad hoc solutions.

### 2.5. Reconstituted Buffer Is as Efficient as PAP to Induce Lipid Peroxidation, Intracellular ROS Formation, Caspase 3/7 Activity and Cell Death

To further support our conclusion that reconstituted buffer (RB) recapitulates most, if not all, of the effects induced by PAP on cells in vitro, we analyzed lipid peroxidation, intracellular ROS formation, caspase 3/7 activation, and cell death. Caspase-3 and Caspase-7 are essential regulators of the apoptotic cascade [79]. To do so, CAL27 and FaDu were incubated for 1 h in RB or PAP. Lipid peroxidation and ROS formation were analyzed immediately after incubation, while caspase 3/7 activation was analyzed 6 h after incubation, and cell death was evaluated 6 h and 72 h post-incubation. Examples of flow cytometry dot plots (forward scatter vs fluorescence for lipid peroxidation) are shown in Appendix A. We found that both treatments are equivalent for all endpoints analyzed (Figure 7). Indeed, in response to either RB or PAP, we observed around a 1.5-fold increase of lipid peroxidation both in CAL27 and FaDu (Figure 7a), around a 3-fold and 2-fold increase of intracellular ROS levels in CAL27 and FaDu, respectively (Figure 7b), around an 8-fold increase in caspase 3/7 activity in CAL27, while no increased activity was detected in FaDu (Figure 7c), and progressive cell death from 6 to 72 h post-treatment, both in CAL27 and FaDu (Figure 7d).

### 2.6. Effect of Direct Plasma Treatment on Lipid Peroxidation, Intracellular ROS Production, Caspase 3/7 Activation and Cancer Cell Viability

In the previous sections, we demonstrated that a reconstituted buffer composed of adequate concentrations of H_2_O_2_ and NO_2_^−^, and adequate pH, is as efficient as the corresponding PAP at triggering cell death by long-lived RONS-dependent processes. As the plasma is composed of ions, electrons, photons, radicals, as well as excited atoms and molecules, and an electromagnetic field, we then focused on the effect of direct plasma treatment on lipid peroxidation, intracellular ROS production, caspase 3/7 activation, and cell viability in CAL27 and FaDu cells to compare it with the effect of indirect plasma treatment. We found that the level of lipid peroxidation was significantly, although moderately, increased after direct plasma treatment compared to indirect plasma treatment (Figure 8a), while the level of intracellular ROS production (Figure 8b) and caspase 3/7 activation (Figure 8c) was similar after both treatments. Despite these moderate effects between direct and indirect plasma treatments, the direct plasma treatment was more efficient than the indirect plasma treatment at inducing cell death in CAL27 and FaDu, insofar as cell viability at 72 h post-treatment was about 28% and 54% in CAL27, and 13% and 20% in FaDu, respectively (Figure 8d). These results suggest that other chemical species and/or plasma physical factors contribute alongside long-lived species to the toxicity of direct plasma treatment.

### 2.7. Contribution of Plasma Treatment Time (Immediate Effects) versus Incubation Time (Early Effects) to the Toxicity of Direct Plasma Treatment

Direct plasma treatment consists of two phases: the period of plasma treatment (immediate effects), which corresponds to the time during which cells covered by PBS are exposed to the plasma, and the period of incubation, which corresponds to the time during which the cells are further incubated in PAP (early effects). Indirect plasma treatment consists only of the second phase. Here, we attempted to better characterize the key drivers of PAP toxicity during direct plasma treatment, and more specifically during the first phase. Foremost, we evaluated the contribution of long-lived species (i.e., H_2_O_2_, NO_2_^−^ and NO_3_^−^) to PAP toxicity during the plasma treatment time (immediate effects). For that, we determined the average concentration of each of these species present in the PBS during the plasma treatment (at a gas flow rate of 1 slm and a treatment distance of 8 mm) from the first second ([H_2_O_2_]_i_ = [NO_2_^−^]_i_ = [NO_3_^−^]_I_ ~0 mM, i standing for initial) to the last second of the 12 min of treatment ([H_2_O_2_]_f_ = 1.6 mM, [NO_2_^−^]_f_ = 0.63 mM, [NO_3_^−^]_f_ = 0.26 mM, f standing for final; data derived from Figure 1). In fact, the concentration of each of these species increases linearly as a function of the plasma treatment time (Figure 9a). Therefore, PBS was treated for 12 min and then CAL27 cells were incubated in this PAP for 6 min, considering that this condition exposes the cancer cells to the same amount of long-lived species as a plasma treatment of 12 min.

In parallel, CAL27 cells were exposed to plasma treatment for 12 min without further incubation time (condition i, immediate effects), to plasma treatment for 12 min followed by 1 h incubation time (condition ii, direct plasma treatment), or to PAP (PBS exposed to 12 min of plasma treatment) for 1 h (condition iii, indirect plasma treatment). Our results showed that the reduction of CAL27 cell viability exposed directly to plasma, followed by 1 h incubation time (condition ii, direct plasma treatment), is the sum of the reduction of cell viability induced only by plasma treatment (condition i, immediate effects), and the reduction of cell viability induced only by 1 h incubation in PAP (condition iii, indirect plasma treatment) (Figure 9b). Indeed, the reduction of cell viability is ~75% in condition ii, ~30% in condition i and ~45% in condition iii (Figure 9b). Furthermore, as the reduction of cell viability after 6 min of incubation in PAP is ~10% (Figure 9b), this suggests that long-lived species cannot, on their own, account for the totality of cell death induced during plasma treatment (immediate effects), and that short-lived reactive chemical species and/or physical parameters (such as electric field, high energy photons, heat) can also contribute to the reduction of cell viability during the plasma treatment.

Recently, Bauer et al. showed that the formation of primary singlet delta oxygen (^1^O_2_) through the complex interaction between NO_2_^−^ and H_2_O_2_ resulted in the inactivation of membrane-associated catalase, secondary ^1^O_2_ generation, further catalase inactivation, intracellular glutathione depletion, and intercellular RONS-mediated apoptosis signaling [34,37,80,81,82]. Therefore, we examined the role of ^1^O_2_ in the reduction of viability during 12 min of plasma treatment (immediate effects). To do so, cells were exposed to plasma in PBS containing either L-histidine or sodium azide (NaN_3_), two well-known ^1^O_2_ quenchers/scavengers [83,84,85]. Our results showed that the presence of L-histidine or NaN_3_ did not prevent the loss of cell viability induced by the 12 min plasma treatment (Figure 9c), thus suggesting that singlet delta oxygen does not play a role in the induction of cancer cell death during plasma treatment (immediate effects).

### 2.8. Normal Cells Are very Sensitive to Direct Plasma Treatment

Thus far, our experiments were conducted using two head and neck tumor cell lines. We then studied the sensitivity of normal cells to direct plasma treatment, indirect plasma treatment, and reconstituted buffer. At first, we used primary gingival keratinocytes (PGK), since these cells are non-cancerous counterparts of CAL27 and FaDu tumor cells. To perform such experiments with normal cells, we used a luminescent-based assay (CellTiter-Glo^®^ assay), which is more sensitive than a colorimetric assay (e.g., MTT assay). The luminescent-based assay determines the number of viable cells in culture by quantitating the amount of ATP present, which indicates the presence of metabolically active cells. We found that the percentage of metabolically active cells dropped to 6%, 33%, and 57% 24 h post direct plasma treatment, indirect plasma treatment, and reconstituted buffer, respectively (see Appendix A). Further incubation up to 72 h post-treatment revealed even a higher loss of activity after direct plasma treatment, while it remained unchanged after indirect plasma treatment and reconstituted buffer (see Appendix A). Since these cells were grown in a very specific cell culture medium (see Material and Methods), had a very limited number of cell divisions, and a slow division time, we also conducted the same experiments using two other normal cell lines, 1Br3 and RPE-hTERT (see Material and Methods). Interestingly, we found that these two cell lines are also very sensitive to direct plasma treatment (Figure 10). Indeed, metabolic activity was reduced to 10% (1Br3) and 30% (RPE-hTERT) 6 h post direct plasma treatment, while it was at least 90% in both cell lines after indirect plasma treatment and reconstituted buffer (Figure 10). Longer incubation times up to 72 h post-treatment led to a further, strong reduction of metabolic activity after direct plasma treatment in 1Br3 (metabolic activity ~1%) and RPE-hTERT (metabolic activity ~2%) (Figure 10). In marked contrast, metabolic viability after indirect plasma treatment and reconstituted buffer was almost unchanged from 6 h up to 72 h post-treatment in 1Br3 (from ~90% to ~80%) and moderately decreased in RPE-hTERT (from ~90% to ~60%) (Figure 10). These unexpected results suggest that normal cells (either primary cells or immortalized with hTERT) are relatively insensitive to the cocktail of long-lived RONS, while they are very sensitive to another type of reactive species, likely short-lived species, or to a physical parameter like UV photons, electromagnetic field or heat. As the treatment of PBS with the plasma does not affect its temperature, which remains at around 23 °C, the role of heat in cell death can be ruled out. Moreover, as the UV-C radiation emitted from the plasma measured at a distance of 8 mm from the reactor’s nozzle is lower than 10^−4^ J/m², the contribution of high-energy photons to the toxicity of the plasma should be negligible.

### 2.9. Direct Plasma Treatment Triggers Strong Cell Detachment and Cell Death Few Hours Post-Treatment in RPE-hTERT Cells

To further decipher the precise nature of these chemical and/or physical factors, RPE-hTERT and CAL27 cells were exposed to direct plasma treatment, and treated PBS was either left on the cells for 1 h (immediate and early effects) or immediately removed (immediate effects only). When removed, treated PBS was replaced by a fresh cell culture medium and the cells were incubated for 6 and 24 h. At these time points, images were taken to visualize the effects of plasma treatment on cell morphology (see Figure 11). For RPE-hTERT cells, the most striking finding was observed after plasma treatment followed by 1 h incubation (direct plasma treatment) insofar as a considerable number of cells were already detached at 6 h post-treatment (Figure 11a, direct treatment with incubation), and were generally forming clusters (Figure 11a and Appendix A). Importantly, cell detachment was not observed in the absence of 1 h incubation following plasma treatment (Figure 11a, direct treatment no incubation).

We then quantified, at 6 h post direct plasma treatment, the number of floating cells, and of the remaining adherent cells by trypan blue cell counting. On the one hand, we found that cells in suspension represent 22 ± 8% of the total cells, and among them 59 ± 13% are still alive at this time point (*n* = 8). However, these cells were unable to re-attach into 24-well plates when seeded in fresh medium. On the other hand, we found that remaining adherent cells represent 26 ± 11% of the total cells, with a cell viability of 89 ± 9% (*n* = 8). It is to highlight that the percentage of living cells measured by trypan blue cell counting is in good agreement with the cell’s metabolic activity determined by the luminescent-based assay (see Figure 10b).

Furthermore, since 6 h post direct plasma treatment, the sum of suspension cells plus remaining adherent cells is about 50% of the total cells. This strongly suggests that about 50% of the cells have already gone through cell lysis at this time point, and, therefore, cannot be detected either by cell counting or by flow cytometry (see Appendix A). Finally, we did not observe cell cycle arrests 6 h post direct plasma treatment (see Appendix A).

For CAL27 cells, the most striking observation was a change in cell morphology (more rounded cells) at 6 h post plasma treatment, whether there was 1 h incubation or not (see Figure 11b). However, this change was less apparent at 24 h post-treatment when there was no incubation (Figure 11b, direct treatment no incubation, T24 h). Together, these observations suggest that CAL27 cells better recover than RPE cells from direct plasma treatment.

### 2.10. Transient Reactive Species, Produced in or Transferred to the Liquid Phase during the Plasma Treatment (Immediate Effects), Sensitize Normal but Not Tumor Cells to PAP

Quantification of the metabolic activity of RPE-hTERT and CAL27 cells at 24 h post-treatment showed that plasma treatment (immediate effects, i.e., no incubation time) has a moderate effect on its own on metabolic activity for both RPE-hTERT and CAL27 (~80% of cell viability), and that the incubation time (early effects) is required to further increase cell death, moderately for CAL27 (~60% of cell viability), but more drastically for RPE-hTERT (~20% of cell viability)—compare “Direct” versus “Plasma” in Figure 12. Since PAP treatment also has a moderate effect on metabolic activity for both RPE-hTER and CAL27 (~80% of cell viability; Figure 12), these data strongly suggest that plasma treatment (immediate effects) and incubation time (early effects) have a synergistic effect on the mortality of RPE-hTERT and confirm an additive effect (see also Figure 9) on the mortality of CAL27.

To further investigate the potential that plasma treatment (immediate effects) “sensitizes” normal, but not tumor cells to PAP, cells were sequentially exposed to (1) plasma treatment for 12 min (immediate effects), after which PAP was removed, and subsequently, (2) untreated PBS or PAP was added for 1 h. We found that untreated PBS added immediately after plasma treatment was not (CAL27) or was moderately (RPE-hTERT) toxic to the cells, while the addition of PAP strongly impinged cell viability to a level similar to that observed for each cell line after direct plasma treatment (compare “Direct” and “Plasma + PAP 1 h” in Figure 12). Collectively, these results support our hypothesis that plasma treatment (immediate effects) potentiates the toxic effect of long-lived RONS on normal cells.

### 2.11. Characterization of Transient Reactive Species Present in PAP

Thus far, we cannot exclude that electric field rather than transient reactive species present during plasma treatment are responsible for the effective killing of normal cells after direct plasma treatment or plasma treatment (immediate effects) + PAP (see above Figure 12). To challenge this hypothesis, RPE-hTERT cells were exposed to direct plasma treatment or to indirect plasma treatment for 1h, in which PAP was added to the cells either immediately (T0 min) or 5 min (T5 min), 10 min (T10 min), 20 min (T20 min), 30 min (T30 min), 45 min (T45 min) or >60 min (>T60 min) after plasma treatment. We observed a time-dependent response showing that, if PAP was added immediately (T0 min) after plasma treatment, then the cell viability was close to that resulting from a direct plasma treatment, while if PAP was added 45 min after plasma treatment (T45 min), then the cell viability was similar to that resulting from an indirect plasma treatment with PAP stored for several hours on the bench (>T60 min) (Figure 13). These results demonstrate that transient reactive species (short-lived species), and not the electric field, contribute mostly to the overall toxicity of plasma on normal cells. Furthermore, since the toxic activity of PAP stored for 10 min (T10 min) on the bench is similar to that of the PAP stored for >60 min (>T60 min), this means that the lifetime of the active species is short (<10 min). Based on the results shown in Figure 13, we estimated the lifetime of the active species to be of a few tens of seconds to a few minutes, therefore excluding plasma-derived very short-lived RONS such as free radicals or singlet delta oxygen (^1^O_2_). We also excluded the role of ^1^O_2_ derived through the interaction of long-lived species in PAP [37] since L-histidine, a quencher of ^1^O_2_ [86], was unable to prevent plasma-induced cell death (Figure 13).

Among the reactive species, ozone (O_3_) and chlorine species (Cl_2_^−^, OCl^−^) are good candidates since they are both generated by cold atmospheric plasmas and their lifetime in liquids is in the range of seconds to hours, depending on the nature of the liquid [87,88,89]. We attempted to detect ozone production in PAP by two spectroscopic methods: directly by its UV absorption at 258–260 nm [90] and indirectly by the absorption detection at 600 nm of the decolorization of potassium indigo trisulfonate dye by ozone [91]. Both methods failed since the high production of H_2_O_2_ and NO_2_^−^ also generates a peak at 258–260 nm and these two compounds together react with potassium indigo trisulfonate. Since hydroxyl radicals are the main oxidants formed in the decomposition of ozone in water [92], we used dimethylsulfoxide (DMSO) to trap hydroxyl radicals [93,94]. Adding DMSO to PAP immediately after plasma treatment strongly prevented cell death (Figure 13). However, insofar as DMSO is also oxidized by hypochlorite [95,96], we used taurine, a well-known hypochlorite scavenger [97,98], to assess the role of this species in plasma-induced cell death. As shown in Figure 13, adding taurine immediately after plasma treatment also strongly prevented cell death, but not to the same extent as with DMSO. Collectively, these results demonstrate that reactive species other than long-lived reactive species (likely ^•^OH through O_3_ decomposition and/or OCl^−^) are produced by CAP and are toxic to normal cells.

## 3. Discussion

It is still currently unclear whether the chemistry of a plasma-activated liquid drives most of the anti-cancer properties of plasma treatment. In an attempt to answer this question, we investigated the difference between direct plasma treatment (which supports the effects of both physical and chemical parameters of plasma) and indirect plasma treatment (which supports only the effects of some of the chemical parameters of plasma) by a CAP, under different experimental conditions (gas flow rate of 0.5 and 1 slm, and treatment distance of 8 and 20 mm), on the cell viability of two cancer cell lines (CAL27 and FaDu).

Several studies have shown that the reactive oxygen and nitrogen species (RONS) play a major role in the plasma anti-cancer properties (reviewed in References [32,99,100]). By measuring the concentration of these species (H_2_O_2_, NO_2_^−^ and NO_3_^−^) as a function of the gas flow rate and treatment distance, we were able to establish a correlation between the concentration of those species in PBS and cell viability. First, we observed that the strongest cell death was obtained when both cell lines were treated at the treatment distance of 8 mm and the gas flow rate of 0.5 slm, a condition that correlates with the highest concentration of both H_2_O_2_ and NO_2_^−^ in PAP. Secondly, by increasing the gas flow rate from 0.5 to 1 slm, while keeping the same treatment distance of 8 mm, the concentration of H_2_O_2_ in PAP did not significantly change, but the concentration of NO_2_^−^ significantly decreased, which correlated with lower cell mortality. Thirdly, by increasing the treatment distance from 8 to 20 mm, and irrespective of the gas flow rate, the concentration of H_2_O_2_ and NO_2_^−^ dropped drastically, correlating with higher cell survival. These results can be summarized as follows in terms of cell death induced by PAP:H_2_O_2_^high^ + NO_2_^high^ > H_2_O_2_^high^ + NO_2_^low^ > H_2_O_2_^low^ + NO_2_^low^

Indeed, we demonstrated in this study that the survival rate of both cell lines is lower at 1.6 mM H_2_O_2_ than 0.8 mM H_2_O_2_, and the addition of NO_2_^−^ to these solutions further decreases cell viability. Furthermore, the dismutation of H_2_O_2_ by catalase completely prevented PAP-induced cell death, confirming the central role of H_2_O_2_ [41,101,102,103]. We also demonstrated that acidification of PAP, although minor (–0.55 at most), is also an important parameter, as recently reported [104]. Collectively, these results have several implications that should be considered to optimize the anti-cancer effects of PAP. It is not only the concentration of H_2_O_2_ that is important, but also the cumulative concentration of H_2_O_2_ + NO_2_^−^ and the pH. Some studies have used H_2_O_2_ in standalone and at concentrations found in plasma-treated liquid as a control to investigate the effect of H_2_O_2_ produced by plasma in solution [41,72,101,102,105,106]. We suggest that, if a plasma-treated solution such as PBS is used to demonstrate its therapeutic potency, it is then important to use an adequate combination of H_2_O_2_ + NO_2_^−^ + pH as control. Indeed, we demonstrated that, in vitro, ad hoc reconstituted solutions are as efficient as PAP to induce lipid peroxidation, intracellular ROS production, caspase activation, and cell death. Recently, it has been shown that PAP can promote an immunogenic phenotype [103,107,108,109,110]. This phenotype is characterized by the expression and release of Damage-Associated Molecular Patterns (DAMPs) from damaged cells leading to specific activation of the immune system through the activation of dendritic cells (DCs) and lymphocytes T CD8+ cytotoxic [111]. As such, cells undergoing immunogenic cell death express calreticulin (CRT), heat shock proteins HSP70 and HSP90, and release in the extracellular environment High Mobility Group Box 1 (HMGB1) protein and ATP. In the future, it would be very interesting to know if a reconstituted buffer composed of H_2_O_2_/NO_2_^−^/acidic pH could also induce immunogenic cell death to a similar extent as the corresponding PAP. Indeed, if a reconstituted buffer can mimic all of the phenotypes induced by indirect plasma treatment (PAP alone), one can wonder of the interest of using plasma to generate RONS in saline solutions, while this can be easily and directly achieved by mixing appropriate concentrations of H_2_O_2_ and NO_2_^−^, and by adjusting the pH.

Unlike indirect plasma treatment, the anti-cancer capacity of PAP using direct plasma treatment includes two phases: the plasma treatment time (phase I, plasma ON, immediate effects) and the incubation time in PAP (phase II, plasma OFF, early effects). In this study, the plasma treatment time was set at 12 min, and the incubation time at 1 h in most of the experiments. We showed that the sensitivity of tumor cells to direct plasma treatment is the additive effect of the plasma toxicity during the plasma treatment time (immediate effects) and of the PAP toxicity during the incubation time in PAP (early effects), making direct plasma treatment more efficient than indirect plasma treatment. Moreover, since cell death induced during plasma treatment (immediate effects) cannot be attributed exclusively to the long-lived species present in PAP, we also had to consider the role of short-lived radicals and atomic species (e.g., O, ^•^NO, ^•^OH, and O_2_^•−^) as well as of non-radical chemical compounds, such as singlet delta oxygen ^1^O_2_ [31], and physical parameters such as UV photons and electric field. In fact, the study of the effects of plasma treatment (immediate effects) on normal cells ruled out a significant action of UV and electric field and brought unexpected results with regards to the contribution of short/intermediate-lived species created during the plasma treatment (immediate effects) and persistent in activated PBS from a few seconds to a few minutes after the plasma is switched OFF (early effects).

Indeed, we demonstrated that normal cells are relatively resistant to reconstituted buffer and indirect plasma treatment (phase II), i.e., to a mixture of H_2_O_2_ + NO_2_^−^ + NO_3_^−^ + acidic pH, and also to plasma treatment (phase I, immediate effects). Taken separately, each phase has little effect on the viability of normal cells, while direct plasma treatment (phases I and II) is very toxic to normal cells. These results reveal that these two phases act in synergy on the death of normal cells (unlike in the case of tumor cells, where the effect of these two phases is additive). Furthermore, the cell death process is extremely fast, since, at 6 h post direct plasma treatment, most of the normal cells, but not of the tumor cells, have already died. These observations paved the way to numerous questions: why are normal cells resistant to indirect plasma treatment, but not tumor cells? Why is there such a big difference in the response to direct plasma treatment between normal and tumor cells? Why is there an additive effect of plasma treatment (immediate effects) and incubation in PAP (early effects) in tumor cells and a synergistic effect of those treatments in normal cells? Additionally, what reactive species contribute to this phenomenon?

It is well described that normal and tumor cells have different redox balances. Many reports have shown that tumor cells counterbalance intrinsic oxidative stress by upregulating antioxidant defense [112,113]. This increase in basal ROS generation renders cancer cells highly dependent on antioxidant systems and more vulnerable to agents that abrogate the antioxidant system and/or increase the ROS level [99,114]. Furthermore, it has also been proposed that tumor cells express more aquaporins at their cytoplasmic membranes than homologous normal tissues, speeding up the uptake of H_2_O_2_ inside their cytosol [115,116]. These features of cancer cells could explain the sensitivity of tumor cells, but not that of normal cells, to indirect plasma treatment or reconstituted buffer, whose toxicity relies essentially on H_2_O_2_ [41]. Recently, Bauer et al. have demonstrated that singlet delta oxygen, ^1^O_2_, which is generated through the interaction between CAP-derived H_2_O_2_ and NO_2_^−^, is at the origin of a cascade reaction in tumor cells, but not in normal cells, leading to inactivation of membrane-associated catalase, which in turns favors H_2_O_2_ uptake and triggers cell death [37,82]. This model could in fact fully explain reconstituted buffer being as effective as indirect plasma treatment regarding the biological response of CAL27 and FaDu to these two treatment modalities, even if this model has been very recently questioned [117].

The striking finding in our work comes from the hypersensitivity of normal cells to direct plasma treatment that cannot be explained only by long-lived reactive species, but that also requires the participation of short/intermediate-lived reactive species. The synergy that we observed between phase I (plasma ON) and phase II (plasma OFF, incubation in PAP) on the death of normal cells strongly suggests that, during phase I, normal cells, but not tumor cells, are “activated” rendering them very sensitive to phase II. Indeed, we demonstrated that normal cells exposed to plasma treatment (phase I only, immediate effects) followed immediately by incubation in another plasma-activated PBS (indirect plasma treatment) exhibit similar sensitivity as if they have been exposed to direct plasma treatment (phase I + phase II). Interestingly, the cell activation phenomenon was previously described by Yan et al. to explain why tumor cells are more sensitive to direct plasma treatment than indirect plasma treatment [63]. In our study, we propose that this activation state plays a minor role in the vulnerability of tumor cells to direct plasma treatment, since the toxicity of phase I and phase II is additive rather than synergic, and although direct plasma treatment remains more efficient than indirect plasma treatment, this difference is less pronounced in tumor cells than in normal cells. Yan et al. showed that, in the tumor pancreatic adeno carcinoma cell line PA-TU-8988T, the desensitization of activation takes 5 h after the CAP treatment [63]. It is possible that, in our experimental conditions, the deactivation state is extremely fast in tumor cells compared to normal cells.

We showed that this “activation state” is triggered by short/intermediate-lived reactive species present in the liquid phase, even after the plasma is switched OFF. The liquid chemistry induced by cold atmospheric pressure plasma is rather complex [31,35,89]. Here, we showed that DMSO (^•^OH and OCl^−^ scavenger) or Taurine (OCl^−^ scavenger) can prevent completely (DMSO) or partially (Taurine) plasma-induced “activation state” in normal cells, suggesting that both chlorine species and hydroxyl radicals are the major reactive species that induce the “activation state”. However, knowing that the lifetime of hydroxyl radicals in solution is of few µs [118], this means that, to interact with the cells, they have to be produced in their vicinity, originating from a more stable species. We propose that this more stable species is ozone (O_3_). Ozone has a lifetime in water at 25 °C and pH 7 of about 200 s [119], but this value depends on several factors including the chemical composition of the solution [88]. Ozone is produced during plasma treatment (phase I, plasma ON, immediate effects) [38,120,121], and can undergo decomposition via a chain reaction mechanism resulting in the production of free hydroxyl radicals (^•^OH) [88]. Unfortunately, we have not been able to measure O_3_ in PAP (by absorption spectroscopy in solution) due to interference with the high concentration of H_2_O_2_ and NO_2_^−^ produced in our experimental conditions. Therefore, further experiments need to be set to validate this hypothesis.

It remains unclear how short/intermediate and long-lived reactive species cooperate to induce cell death preferentially in normal cells over tumor cells. It is important to keep in mind that the cellular membrane is the first line of defense towards exogenous stress, such as chemically reactive species, and it is well documented that the membrane composition is different between normal and tumor cells [122,123,124]. Membrane composition is essential for cell survival as lipids are used for energy storage, compartmentalization and signaling [125]. We propose that lipids undergo oxidation upon plasma treatment, leading to lipid peroxidation, which in turn increases membrane permeability [126,127,128], likely due to transient pore formation [128], thus allowing for more extracellular compounds, such as long-lived RONS, to enter into the cells. Moreover, since lipid peroxidation increases membrane fluidity and decreases the electric field threshold needed for transient pore formation [127], we cannot rule out a role, albeit modest, of the electric field in the toxicity of direct plasma treatment, especially towards normal cells.

In conclusion, we have shown that normal and tumor cells respond differently to plasma-activated liquid, whether we consider the direct or indirect plasma treatments. Indeed, we described, in this study, a synergistic effect of short/intermediate-lived species and long-lived species to trigger important cell death in normal cells, while these cells are relatively resistant to long-lived species alone. In contrast, tumor cells are both sensitive to direct and indirect plasma treatments, essentially due to the action of long-lived species. While some models have been proposed to explain the anti-cancer specificity of plasma-activated liquids [37,115], it remains important to determine the precise role of the cell membrane and its composition in the cell sensitivity or cell resistance to plasma treatment.

## 4. Materials and Methods

### 4.1. Cell Culture

FaDu (ATCC^®^ HTB-43™) and CAL27 (ATCC^®^ CRL-2095™) are two human squamous cell carcinomas derived from the pharynx and the tongue, respectively, and were provided by Professor Toillon (Lille University, Lille, France). 1BR3 cells are human primary skin fibroblasts [129] and were provided by Professor Penny Jeggo (GDSC, Brighton, UK). Primary skin fibroblasts were used at passages 16 to 18. Squamous cell carcinomas and fibroblasts were cultured in Dulbecco’s modified Eagle Medium (DMEM, Invitrogen™ Life Technologies™, Saint Aubin, France) containing 4.5 g/L glucose, 2 mM L-glutamine (Invitrogen™ Life Technologies™), 100 U/mL-100µg/mL penicillin-streptomycin (Invitrogen™ Life Technologies™), 1 mM Non-Essential Amino Acids (Invitrogen™ Life Technologies™) and 10% fetal calf serum (Eurobio Scientific, Les Ulis, France). Primary Gingival Keratinocytes (ATCC^®^ PCS-200-014™) were grown in Dermal Cell Basal Medium (ATCC^®^ PCS-200-030™), supplemented with Keratinocyte Growth Kit (ATCC^®^ PCS-200-040™), and were used at passage 2. hTERT RPE-1 (ATCC^®^ CRL-4000™) are epithelial cells immortalized with hTERT and were provided by Dr. Arturo Londono (Institut Curie, Paris, France). These cells were cultured in DMEM/F12 GlutaMAX™ containing 100 U/mL-100µg/mL penicillin-streptomycin (Invitrogen™ Life Technologies™), and 10% fetal calf serum (Eurobio Scientific). All cells were grown at 37 °C, 5% CO_2_ in a humidified atmosphere, and were regularly checked for mycoplasma contamination using Venor^®^GeM Advance Mycoplasma Detection Kit (Biovalley, Nanterre, France). Authentication of FaDu and CAL27 cell lines was performed by short tandem repeat (STR) analysis using 9 STR markers (D5S818, D13S317, D7S820, D16S539, VWA, TH01, AM, TPOX, and CSF1PO). Cells were detached using trypsin–EDTA 0.05% (Invitrogen™ Life Technologies™).

### 4.2. Plasma Device and Operative Conditions

The conceptual view of the plasma reactor is depicted in Figure 14. It consists of a stainless-steel needle (0.7 mm inner and 1.4 mm outer diameters), inserted inside a dielectric tube made of quartz, and biased electrically by applying high voltage square positive pulses (amplitude of 6 kV, pulse width of 4.8 μs, rise and fall times of around 25 ns) at a repetition rate of 10 kHz using a commercial power supply (DEI PVX-4110). The distance between the needle’s tip and the reactor’s nozzle is fixed (55 mm). The ground electrode (10 mm width), made of copper, is wrapped around the dielectric tube, centered at the tip of the needle. Helium (Alphagaz 2 He type S11, Air Liquide, Corbeil-Essonnes, France) with an admixture of 0.2% of molecular oxygen (Alphagaz 1 O_2_ type L50, Air Liquide) is injected through the needle at a flow rate of either 0.5 or 1 slm, regulated by flowmeters (GF40-SA46 (for He) and SLA5800 (for O_2_), Brooks instrument, Serv’Instrumentation, Irigny, France). Thus, a dielectric barrier discharge (DBD) is formed, and a plasma jet propagates outside the quartz tube through the He channel into the surrounding ambient air. The distance between the tube orifice and the surface of the liquid is fixed at 8 or 20 mm.

The micro-plasma jet reactor was placed vertically with the gas flowing downwards for interaction with the PBS covering the different cellular models adhered to the bottom of the plate wells. The plasma propagated through the capillary tube and the He channel into the surrounding atmosphere, and either the plasma or its gaseous effluent entered the buffer solutions with a small admixture of the surrounding air.

### 4.3. Plasma Treatment

In short-term in vitro experiments (incubation up to 24 h post-treatment), 5 × 10^4^ to 2 × 10^5^ cells were seeded per well in 24-well plates in 1 mL of complete medium and incubated for 24 to 72 h so that the cells were between 50% and 60% confluent at the time of plasma treatment. In long-term in vitro experiments (incubation from 48 h to 72 h post-treatment), 2 × 10^4^ to 10^5^ cells were seeded per well in 24-well plates in 1 mL of complete medium and incubated for 24 h. For direct plasma treatment, the cell culture medium was removed, the cells washed one time with phosphate-buffered saline containing 0.9 mM CaCl_2_ and 0.49 mM MgCl_2_ (named only PBS for convenience in this manuscript), and 800 µL of PBS were added to each well. PBS covering the cells was then exposed to He/O_2_ plasma (experimental conditions described in the previous section), resulting in plasma-activated PBS (named PAP for convenience in this manuscript). At the end of the plasma treatment (immediate effects), PAP was either left in contact with cells at room temperature for 1 h (early effects) and then removed (direct plasma treatment) or immediately removed (only immediate effects). For indirect plasma treatment, 800 µL of PBS were added per empty well and treated with He/O_2_ plasma, resulting also in PAP. Thereafter, PAP was put in contact with the cells for 1 h, after which it was removed. For reconstituted buffer, 800 µL of untreated PBS containing the same concentrations of H_2_O_2_, NO_2_^−^ and NO_3_^−^ and the same value of pH as in plasma-treated (for 12 min) PBS was put in contact with the cells for 1 h, after which it was removed. In all cases (direct and indirect plasma treatments, as well as reconstituted buffer), after removal of PAP, 1 mL of complete medium was added to the cells, and the plates incubated at 37 °C and 5% CO_2_ in a humidified atmosphere for from 6 h to 72 h.

### 4.4. Cell Viability Assays

The cell viability was determined using MTT, CellTiterGlo^®^, and trypan blue assays. MTT assay is a colorimetric assay and CellTiterGlo^®^ is a luminescent assay, both assessing the metabolic activity of the cells that is proportional to the number of viable cells under cytotoxic conditions. For the MTT assay, the cell culture medium was removed, and the cells covered with 500 µL of complete medium containing 0.5 mg/mL thiazolyl blue tetrazolium bromide (MTT) (Sigma-Aldrich, Saint-Quentin Fallavier, France). The plates were incubated 3–4 h at 37 °C until a purple precipitate was visible. The resulting intracellular purple formazan was then solubilized by adding 500 µL of isopropanol 95%/hydrochloric acid 1.5%. Spectrophotometric quantification was then performed at 570 nm using a well plate reader (Infinite^®^ M200 PRO Tecan, Lyon, France). For the CellTiterGlo^®^ assay, the cell culture medium was removed, and the cells covered with 300 µL of a solution 1/1 (vol/vol) of complete culture medium/CellTiterGlo^®^ reagent. The plates were shaken for 15–20 min to allow complete lysis of cells and 100 µL of solution were transferred into 96-well white plates. Luminescence signal was recorded on a plate reader (VICTOR™X3, Perkin Elmer, Villebon-sur-Yvette, France). The trypan blue assay is a direct counting of living cells, which exclude the dye, and of dead cells, which incorporate the dye. To perform the assay, 6 h post-treatment, cell culture medium containing floating cells was transferred into 2 mL Eppendorf tubes, the tubes centrifuged for 5 min at 6000 rpm, after which the medium was discarded. 100 µL of fresh medium was added to each tube, and the cells were counted. In parallel, remaining adherent cells in each well were detached by trypsin, collected in 2 mL Eppendorf tubes and further processed, as described above. Cell counting was performed on a LUNA-II automatic cell counter (Logos Biosystem, Villeneuve d’Ascq, France).

### 4.5. Quantification of Hydrogen Peroxide (H_2_O_2_), Nitrite (NO_2_^−^), and Nitrate (NO_3_^−^)

The concentration of hydrogen peroxide in PBS or PAP was determined using titanium oxysulfate, as previously described [46]. The absorbance was measured at 407 nm using a well plate reader (Infinite^®^ M200 PRO Tecan). All reagents used for the H_2_O_2_ quantification were from Sigma-Aldrich. The quantification of nitrite and nitrate was performed using Cayman’s nitrate/nitrite colorimetric assay kit (Interchim, Montluçon, France) according to the supplier’s instructions. The absorbance was measured at 540 nm with the same spectrophotometric system.

### 4.6. Dismutation of H_2_O_2_ by Catalase

A 10 mg/mL stock solution of catalase (Sigma-Aldrich, specific activity 2000–5000 units/mg of protein) was prepared in 50 mM potassium phosphate buffer. This solution was diluted 10,000 times in PBS or PAP. Catalase was added to PBS immediately after plasma treatment (for direct plasma treatment) and just before the addition of PAP to the cells (for indirect plasma treatment), so it was present only during the incubation time.

### 4.7. Quenching of Other ROS by DMSO, Taurine, NaN_3_, or L-Histidine

Stock solutions of Taurine, NaN_3_, and L-histidine at 500 mM were prepared in distilled water (Taurine and NaN_3_) and PBS (L-histidine). The final concentration of Taurine was 5 mM, of NaN_3_ 1 mM, and L-histidine 1 or 5 mM. DMSO was used at a final concentration of 1%. Compounds were added to PBS either before or immediately (within 5 s) after plasma treatment. Mock-treated cells were handled the same way as treated cells.

### 4.8. Measurements of pH and Temperature

The pH of PBS or PAP was measured using a SevenEasy™ pH meter S20 fitted with an InLab^®^ Microelectrode (Mettler Toledo, Viroflay, France.). The temperature of PBS or PAP was measured (within 5 s after plasma treatment for PAP) with a thermocouple that was placed inside the solution.

### 4.9. Measure of Lipid Peroxidation and Intracellular ROS Formation

Cells were seeded in 24-well plates and incubated in a humidified incubator at 37 °C/5% CO_2_ for 1 to 3 days depending on the number of seeded cells. When cells were about 80% confluent, the medium was removed and replaced by 0.25 mL of fresh DMEM high glucose, containing 5 µM BODIPY 581/591 C-11 (ref D3861, Invitrogen, Les Ulis, France) to assess for lipid peroxidation, or 5 µM H2DCFDA (ref C6827, Invitrogen) to assess for intracellular ROS production. The plates were further incubated for 30 min, after which the medium was discarded, the cells washed 1× with fresh complete medium, and 1× with PBS. Following 1 h incubation post direct and indirect plasma treatments or reconstituted buffer, the medium was removed, the cells washed 1× with PBS and trypsinized. The cells were collected in FACS tube in 0.4 mL of MACS buffer (PBS containing 2% BSA, 1 mM EDTA, and 0.09% sodium azide, Miltenyi Biotec, Paris, France) and analyzed by flow cytometry (BD LSRFortessa™ X-20, BD, Le Pont de Claix, France), as per the manufacturer’s instructions.

### 4.10. Flow Cytometry Analysis of RPE-hTERT Cells Using Propidium Iodide (PI)

RPE-hTERT cells were exposed to direct plasma treatment (12 min plasma treatment (immediate effects) followed by 1 h incubation (early effects)) and further incubated in a complete cell culture medium for 6 h. All cells were then collected into Falcon tubes. After centrifugation, cell pellets were either resuspended in MACS buffer containing 10 µg/mL of PI or resuspended in PBS and fixed in 70% EtOH. Fixed cells were then centrifuged, washed once in MACS buffer, and resuspended in MACS buffer containing 10 µg/mL of PI. The cells were analyzed by flow cytometry (BD LSRFortessa™ X-20, BD), as per the manufacturer’s instructions.

### 4.11. Apoptosis Assay Using Caspase Detection Method

Caspase 3/7 activity was monitored 6 h after treatment using Caspase-Glo^®^ 3/7 Assay System (Promega, Charbonnières-les-Bains, France).

### 4.12. Statistical Analyses

All the results are expressed as mean ± standard deviation (SD) of 2 to17 independent experiments, with each time point in duplicate. The significance level, or *p*-value, is calculated using the students *t*-test and displayed on the figure plots as NS: *p >* 0.05; *: *p ≤* 0.05; **: *p ≤* 0.01; ***: *p ≤* 0.001; ****: *p ≤* 0.0001.

## 5. Conclusions

Plasma-activated liquid seems to be a promising strategy to treat cancer in vivo, either by direct injection of the activated liquid at the tumor sites [53,130,131] or by intraperitoneal administration [103,132,133,134,135]. One of the main challenges is to determine the optimum conditions of plasma treatment so that the treated solution contains all the ingredients for optimal efficacy in vivo (strong effects on tumor cells while sparing healthy tissue). Thus far, RONS produced by plasma, and, among them, long-lived species like H_2_O_2_, are pointed out as the main drivers of the cellular response. In this study, we found, using PBS as the recipient of plasma treatment, that an appropriate combination of H_2_O_2_, NO_2_^−^ and acidic pH reproduces the cell toxicity observed in vitro with PAP. Therefore, one can ask what is the advantage of using cold atmospheric plasmas to activate a solution if this one can be obtained by mixing solutions of H_2_O_2_, NO_2_^−^ and adjusting the pH?

However, we found that direct plasma treatment is more effective than indirect plasma treatment in vitro, especially towards normal cells, likely because of the additional contribution of short-lived species [62], which can operate during the plasma treatment (immediate effects) and the first seconds/minutes of incubation in PAP after the plasma being switched OFF (early effects). However, this response is also intrinsically linked to the composition of the treated solution. Indeed, if these highly reactive short-lived species are quenched in the liquid during plasma treatment, and despite the formation of long-lived species, the cell response will be different. This is what might occur when using a complex liquid, like cell culture medium, that contains several factors (e.g., amino acids, vitamins, serum) that can act as quenchers of the short-lived species.

Nevertheless, it becomes more challenging to treat cancer cells in vivo with this modus operandi. Indeed, in all reported in vitro studies, immediately after plasma treatment, cells were left in the plasma-activated solution for several minutes to several hours (see for example Reference [136]) to allow essentially long-lived reactive species to interact efficiently with the cells. In vivo, it is not easy to figure out how to cover tumor cells with a liquid phase constrained in space, expose it to plasma, and then left it in place to ensure an efficient anti-cancer response. 

To conclude, the large majority of in vitro studies, including ours, have been conducted in the presence of an excess of liquid phase (e.g., buffered solution, cell culture medium) with regards to the volume of treated cells. In vivo, this situation is unlikely to occur in the treatment of solid tumors. Even in the first report of clinically treated patients by CAP with locally advanced head and neck cancer, it is hard to visualize the liquid interphase (see Figure 3 in Reference [77]). It is a wet surface rather than a liquid. Moreover, cautions have to be taken in extrapolating from experiments on malignant (or other) cells in culture to the in vivo state [137]. Indeed, intracellular O_2_ levels are low for most cells in the human body [138], which diminishes reactive species formation, while cells in vitro are grown in normoxia. Besides, cell-culture media are often deficient in antioxidants and contain ‘free’ iron ions, present as contaminants or even added deliberately (e.g., iron(III) salts are added to DMEM (Dulbecco’s modified Eagle’s medium)), which may further catalyze reactive species, such as hydroxyl radical [139], in the medium. Therefore, we believe that to better translate in vitro studies into in vivo studies, more relevant in vitro models should be used to better mimic the in vivo situation. 

## Figures and Tables

**Figure 1 cancers-13-00615-f001:**
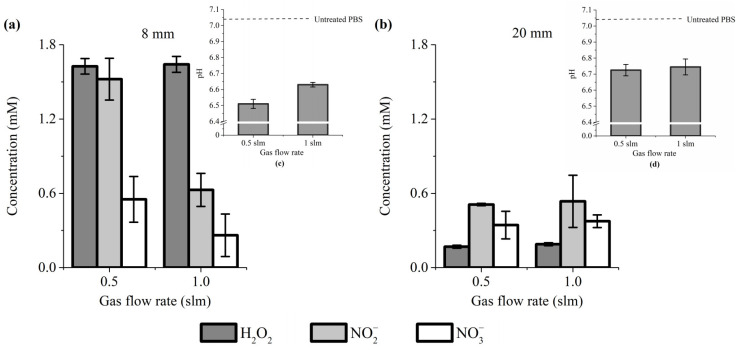
H_2_O_2_, NO_2_^−^ and NO_3_^−^ concentration and pH value (insets) change in PAP. Plasma treatment of 12 min, at gas flow rates of 0.5 and 1 slm (99.8% He/0.2% O_2_), and treatment distances of 8 mm (**a**) and 20 mm (**b**). The data shown are the mean ± SD of 3 (**a**) and 2 (**b**) independent experiments.

**Figure 2 cancers-13-00615-f002:**
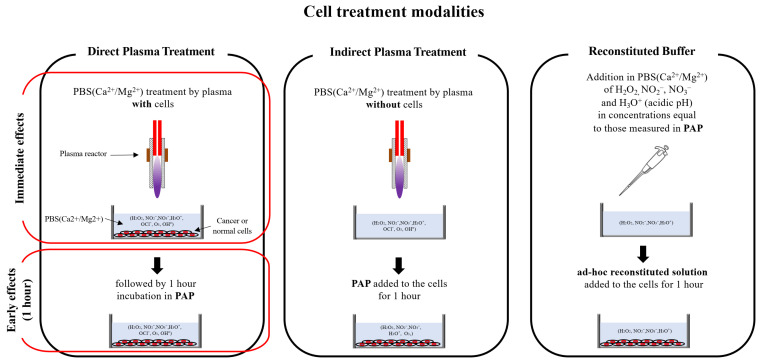
Schematic illustration of the different modalities used for cell treatment.

**Figure 3 cancers-13-00615-f003:**
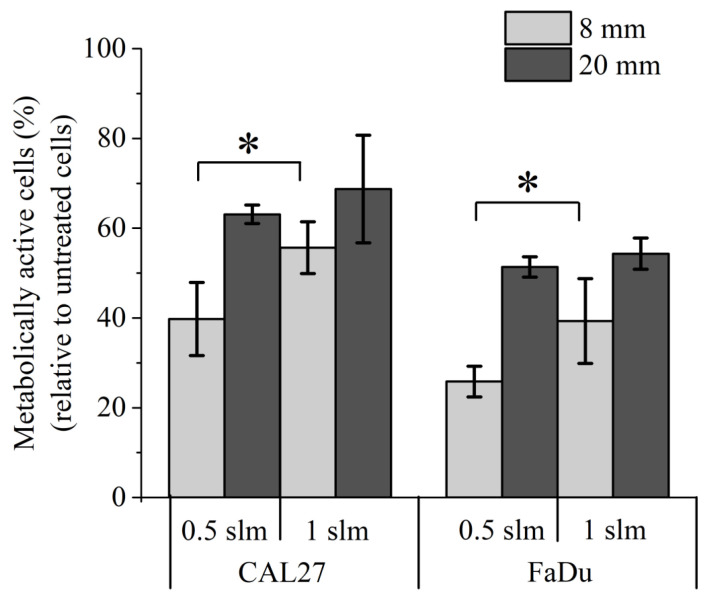
Effect of PAP on the viability of CAL27 and FaDu. Treatments performed for 12 min at distances of 8 and 20 mm, and at gas flow rates of 0.5 and 1 slm (99.8% He/0.2% O_2_). Cell viability was monitored 24 h post-treatment by MTT assay. The data are the mean ± SD of 3 (0.5 slm) or 6 (1 slm) independent experiments. Statistical significance NS: *p >* 0.05; *: *p ≤* 0.05; (*t*-test).

**Figure 4 cancers-13-00615-f004:**
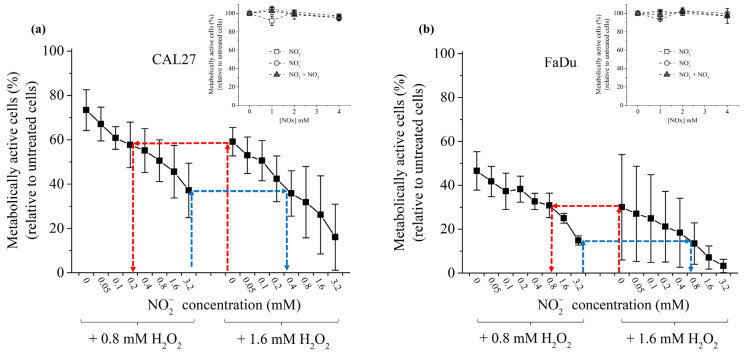
Synergistic effect of H_2_O_2_ and NO_2_^−^ in inducing cancer cell death. CAL27 (**a**) and FaDu (**b**) were incubated in PBS containing H_2_O_2_ at 0.8 or 1.6 mM and increasing concentrations of NO_2_^−^ from 0 to 3.2 mM. The red and blue dashed arrows highlight the fact that the same % of cell viability can be reached using the appropriate combination of H_2_O_2_ + NO_2_^−^. Insets: cell viability of CAL27 (**a**) and FaDu (**b**) in the presence of increasing concentrations of NO_2_^−^, NO_3_^−^ and NO_2_^−^ + NO_3_^−^. The data are the mean ± SD of 2 to 4 independent experiments.

**Figure 5 cancers-13-00615-f005:**
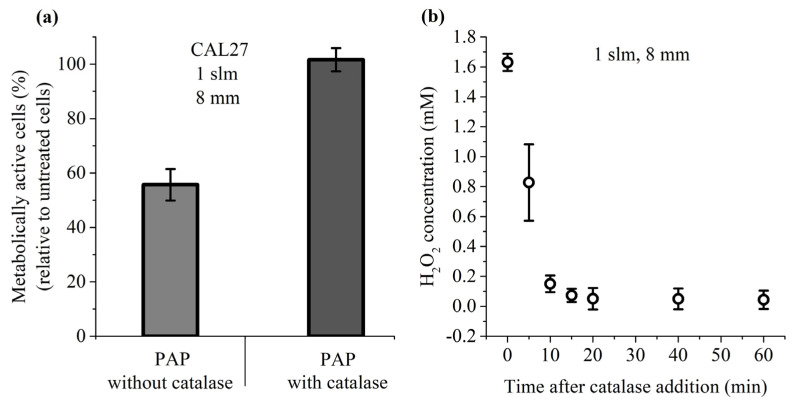
Catalase prevents the toxicity of PAP. (**a**) CAL27 cells were incubated for 1 h in PAP alone or PAP with catalase. Catalase was added after plasma treatment. Cell viability was monitored 24 h post-treatment. (**b**) Concentration of H_2_O_2_ in PAP as a function of the time elapsed after the addition of catalase. The data are the mean ± SD of 3 to 6 independent experiments.

**Figure 6 cancers-13-00615-f006:**
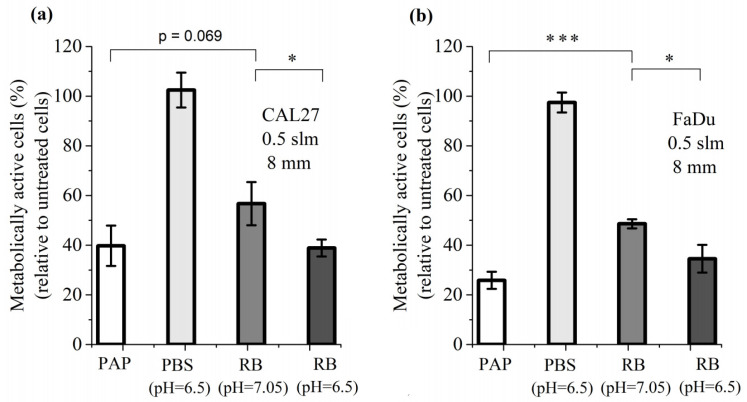
Combined effect of pH and RONS on the cell viability of CAL27 (**a**) and FaDu (**b**) cell lines. Cell viability was assessed 24 h after 1 h incubation in PAP, in PBS pH 6.5, and in reconstituted buffer (RB) pH 7.05 or pH 6.5. RB consists of 1.6 mM H_2_O_2_, 1.5 mM NO_2_^−^ and 0.55 mM NO_3_^−^. The results presented here correspond to a gas flow rate of 0.5 slm (99.8% He/0.2% O_2_), a treatment distance of 8 mm and a treatment time of 12 min. The data are the mean ± SD of 3 independent experiments. Statistical significance NS: *p >* 0.05; *: *p ≤* 0.05; ***: *p ≤* 0.001 (*t*-test).

**Figure 7 cancers-13-00615-f007:**
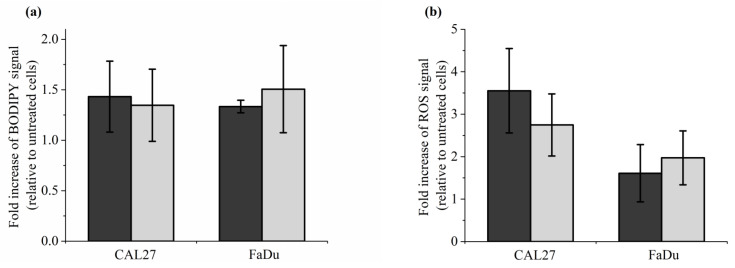
Reconstituted buffer (RB) and PAP are equivalent in inducing lipid peroxidation, an intracellular increase of ROS, caspase 3/7 activation, and cell death. CAL27 and FaDu cells were exposed to PAP (indirect plasma treatment) or RB for 1 h, and then the cells were either immediately collected for evaluation of lipid peroxidation (**a**) and intracellular ROS formation (**b**), or further incubated in a fresh cell culture medium for 6 h to evaluate caspase 3/7 activation (**c**) or for 6 h, 24 h and 72 h to evaluate cell death (**d**). The results presented here correspond to a gas flow rate of 1 slm (99.8% He/0.2% O_2_), a treatment distance of 8 mm, and a treatment time of 12 min. The data are the mean ± SD of 4 to 7 independent experiments for (**a**,**b**), 2 independent experiments for (**c**), and 3 to 4 independent experiments for (**d**).

**Figure 8 cancers-13-00615-f008:**
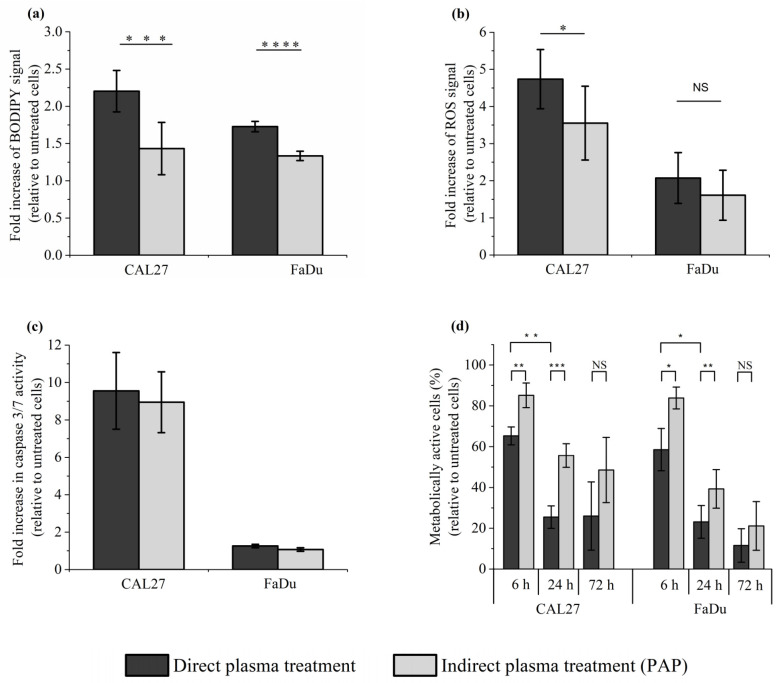
Direct plasma treatment is slightly more efficient than indirect plasma treatment (PAP) at inducing lipid peroxidation, an intracellular increase of ROS, and cancer cell death, while no significant difference was observed between these two treatment methods at inducing caspase 3/7 activation. CAL27 and FaDu cells were exposed to direct and indirect (PAP) plasma treatments, and then the cells were either immediately collected for the evaluation of lipid peroxidation (**a**) and intracellular ROS formation (**b**), or further incubated for 6 h to evaluate caspase 3/7 activation (**c**) or for 6 h, 24 h and 72 h to evaluate cell death (**d**). The results presented here correspond to a gas flow rate of 1 slm (99.8% He/0.2% O_2_), a treatment distance of 8 mm and a treatment time of 12 min. The data are the mean ± SD of 4 to 7 independent experiments for (**a**), 4 independent experiments for (**b**), 2 independent experiments for (**c**), and 3 to 4 independent experiments for (**d**). Statistical significance NS: *p >* 0.05; *: *p ≤* 0.05; **: *p ≤* 0.01; ***: *p ≤* 0.001; ****: *p ≤* 0.0001 (*t*-test).

**Figure 9 cancers-13-00615-f009:**
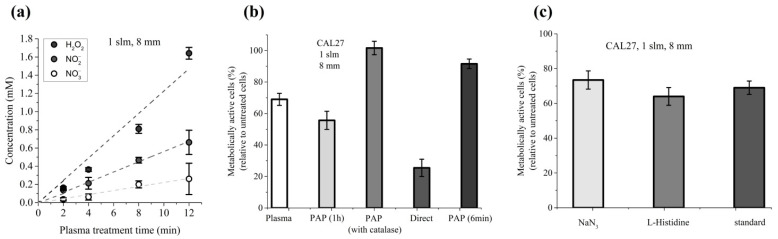
Cumulative contribution of plasma treatment time (immediate effects) and incubation time (early effects) in the reduction of cell viability induced by direct plasma treatment. (**a**) The concentrations of H_2_O_2_, NO_2_^−^, and NO_3_^−^ were measured as a function of the plasma treatment time. (**b**) CAL27 cells were exposed to 12 min of plasma treatment (plasma, immediate effects), to indirect plasma treatment for 1 h without (PAP 1 h) or with catalase (PAP with catalase), for 6 min (PAP 6 min), or to direct plasma treatment (direct). (**c**) CAL27 were exposed to 12 min of plasma treatment in PBS only (standard) or in PBS containing 1 mM L-histidine or 1 mM NaN_3_. Cell viability was measured 24 h post-treatment using MTT assay. The experiments were performed at a gas flow rate of 1 slm (99.8% He/0.2% O_2_) and a treatment distance of 8 mm. The data are the mean ± SD of 3 to 6 independent experiments.

**Figure 10 cancers-13-00615-f010:**
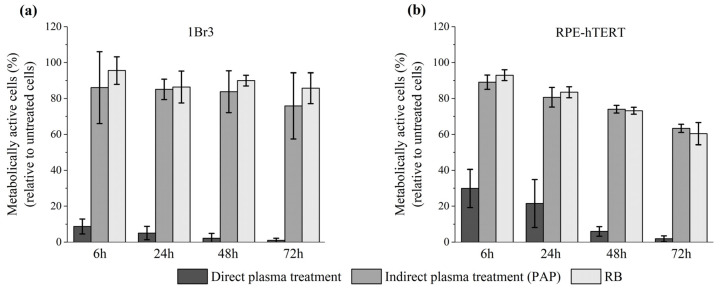
Severe loss of metabolic activity of normal cells after direct plasma treatment. (**a**) Primary human fibroblasts (1Br3) and (**b**) normal epithelial cells (RPE-hTERT) were exposed to direct plasma treatment, indirect plasma treatment (PAP), and reconstituted buffer (RB). At 6 h, 24 h, 48 h, and 72 h post-treatment, metabolic activity was quantified using CellTiter-Glo^®^ luminescent assay. Conditions used were those corresponding to a gas flow rate of 1 slm (99.8% He/0.2% O_2_), a treatment distance of 8 mm, and a treatment time of 12 min. The data are the mean ± SD of 3 to 17 independent experiments.

**Figure 11 cancers-13-00615-f011:**
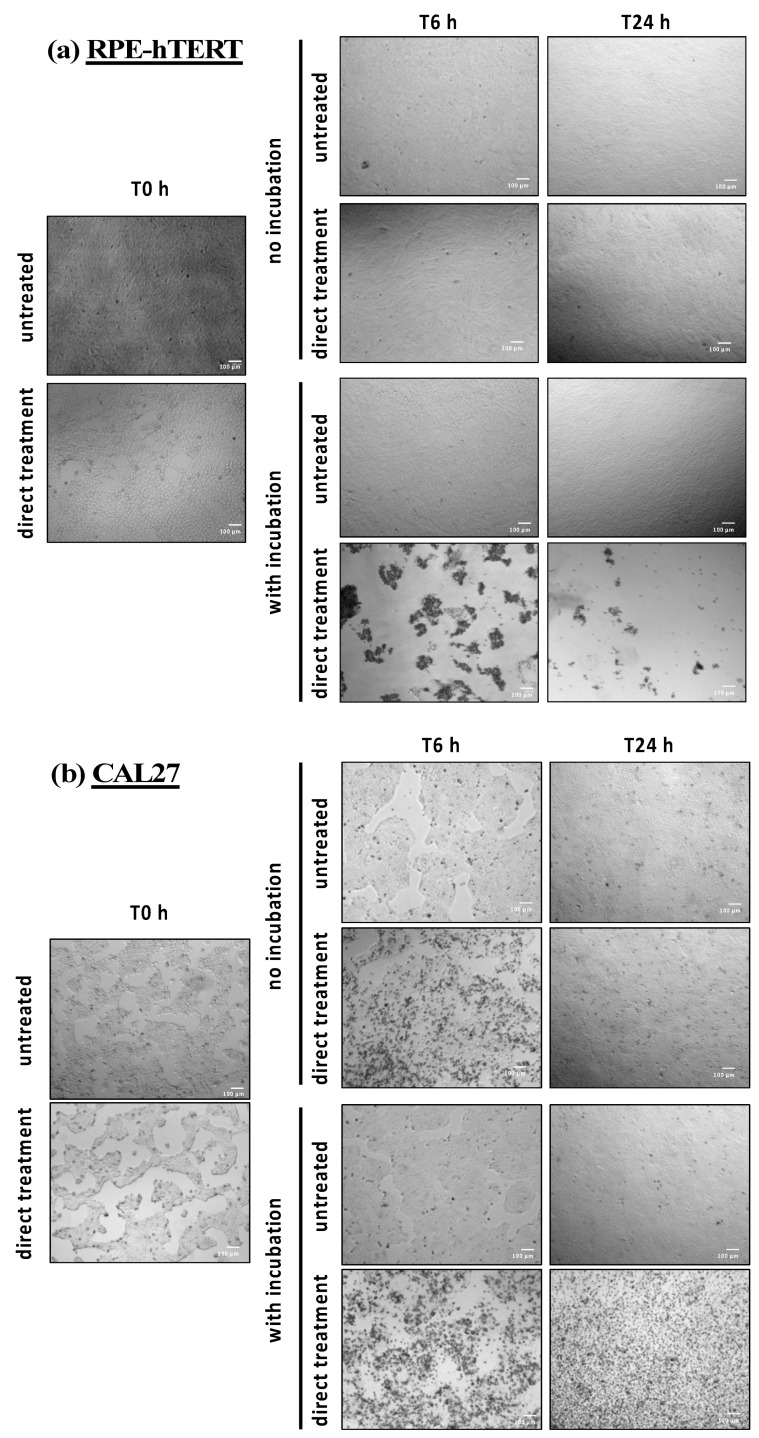
Change in cell shape and cell number after direct plasma treatment. (**a**) RPE-hTERT and (**b**) CAL27 cells were exposed in PBS to plasma treatment and further incubated (with incubation—early effects) or not (no incubation—immediate effects) in PAP. Photographs were taken immediately (T0 h), 6 h (T6 h) or 24 h (T24 h) after plasma treatment, performed at a gas flow rate of 1 slm (99.8% He/0.2% O2), a treatment distance of 8 mm and a treatment time of 12 min. Photographs were recorded on a Celena^®^ S digital imaging system (Logos biosystem, Villeneuve d’Ascq, France) at 4× magnification. Photographs are representative of two independent experiments.

**Figure 12 cancers-13-00615-f012:**
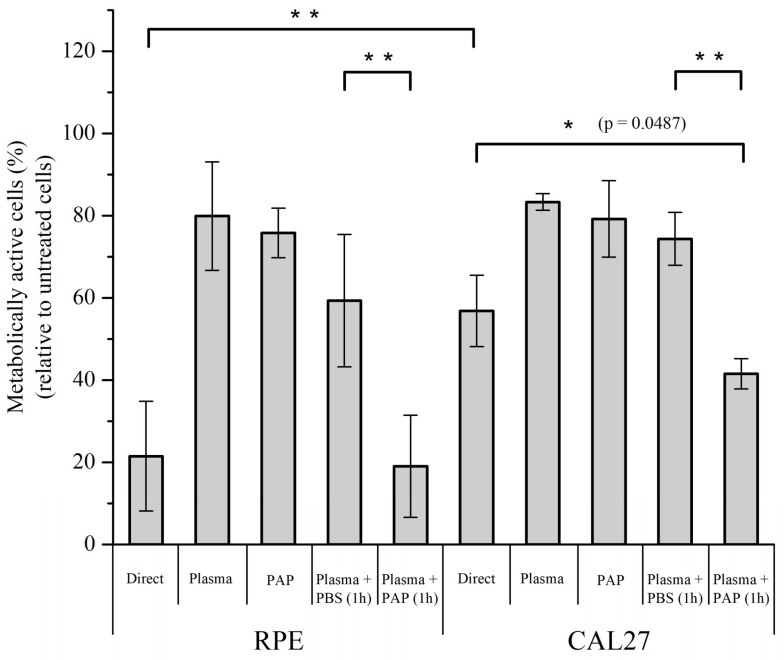
The incubation time (early effects) following plasma treatment (immediate effects) is required to achieve efficient cell killing after direct plasma treatment. RPE-hTERT and CAL27 cells were exposed to plasma treatment for 12 min and PAP was either left on the cells for 1 h (Direct) or immediately removed (Plasma, immediate effects). Alternatively, cells were exposed to plasma-activated PBS for 1 h (PAP), or to plasma treatment for 12 min followed by incubation in untreated PBS (Plasma + PBS) or in another PAP (Plasma + PAP) for 1 h. Metabolic cell activity was quantified using a CellTiter-Glo^®^ luminescent assay 24 h post-treatment. Plasma treatments were performed at a gas flow rate of 1 slm (99.8% He/0.2% O_2_) and a treatment distance of 8 mm. The data are the mean ± SD of 3 to 17 independent experiments. Statistical significance NS: *p >* 0.05; *: *p ≤* 0.05; **: *p ≤* 0.01; (*t*-test).

**Figure 13 cancers-13-00615-f013:**
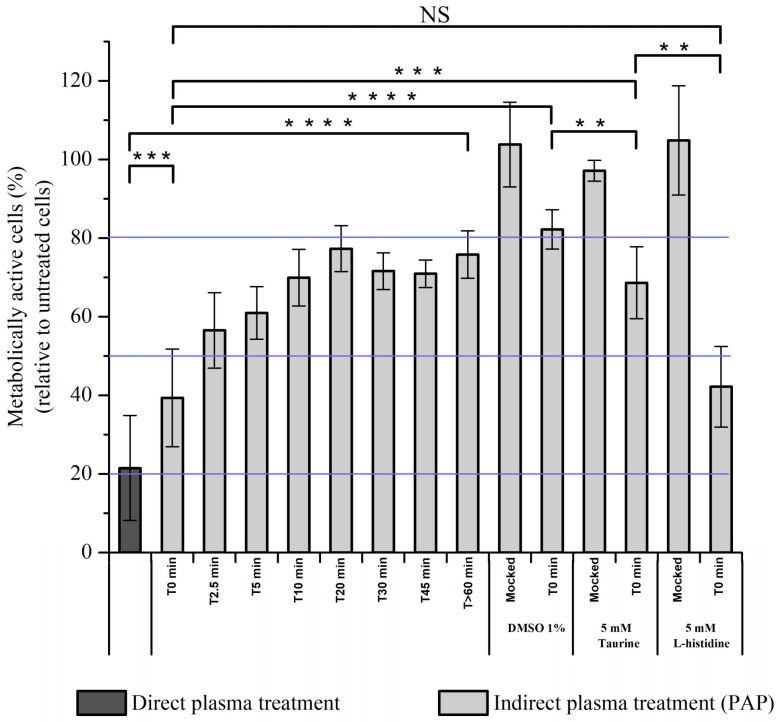
Full PAP toxicity depends on transient reactive species. RPE-hTERT cells were exposed to direct plasma treatment in PBS or PAP (indirect plasma treatment) either not stored (T0 min) or stored for various time periods on the bench (from 0 min to >60 min) after plasma treatment, before being added onto the cells for 1 h. Alternatively, DMSO at a final concentration of 1%, and Taurine and L-histidine at 5 mM were added immediately after plasma treatment to PAP, which was immediately added to the cells for 1 h. Cell viability was quantified at 24 h post-treatment using CellTiter-Glo^®^ luminescent assay. Solid blue lines indicate the average cell viability after direct plasma treatment (around 20%, minimum of cell viability and maximum concentration of transient reactive species), the average cell viability after indirect plasma treatment (PAP, around 80%, maximum of cell viability and no transient reactive species), and the EC50 (around 50%). Plasma treatments were performed at a gas flow rate of 1 slm (99.8% He/0.2% O_2_), a treatment distance of 8 mm, and a treatment time of 12 min. The data are the mean ± SD of 3 to 17 independent experiments. Statistical significance NS: *p >* 0.05; *: *p ≤* 0.05; **: *p ≤* 0.01; ***: *p ≤* 0.001; ****: *p ≤* 0.0001 (*t*-test).

**Figure 14 cancers-13-00615-f014:**
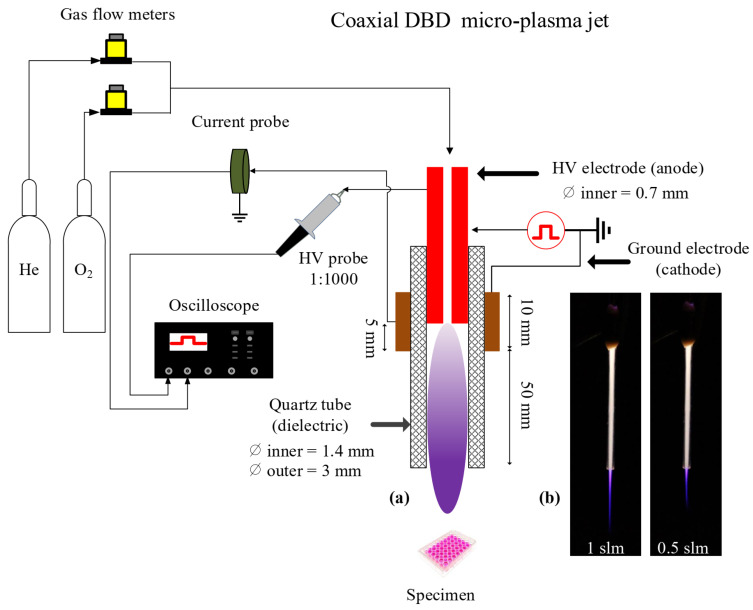
(**a**) Conceptual view of the experimental setup and of the cold atmospheric-pressure plasma jet reactor used in this study. (**b**) Photographs of the home-made plasma jet with the plasma ON, operated in pure He at two different gas flow rates (1 and 0.5 slm), and impinging on the surface of a liquid positioned at 8 mm from the reactor’s nozzle.

## Data Availability

The data presented in this study are available on request from the corresponding author.

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
