# Peer review of "Role of Short- and Long-Lived Reactive Species on the Selectivity and Anti-Cancer Action of Plasma Treatment In Vitro"

_cancers, 2021, doi:10.3390/cancers13040615_

Round 1

Reviewer 1 Report

This paper reports about the potential use of plasmas for cancer treatment. The authors

assessed the anti-tumor capacity of direct and indirect treatments using two models of Head & Neck cancer cells, CAL27 and FaDu. It was showed that direct treatment is more effective. It also provides evidences that during phase II the concentration of RONS in combination with the acidic pH is the main driver of plasma-induced PBS toxicity, while during phase I, other chemical or/and physical parameters are also important.  

It is an interesting paper, contains novel aspects which provides interesting results with appropriate references, therefore it is recommended for publication after a few minor revisions shown below.

  1. The authors should revise the English; there are some grammatical errors.
  2. I don’t understand why the authors divide the abstract in four points. I suggest that should be eliminated.
  3. Some figures should be improved regarding the quality, in particular, figures 1, 3, 7, 8 and 10.
  4. In materials methods section, subsections: Cell culture, Plasma treatment and Cell viability assay, where it is written milliliters or microliters, it should appear mL, instead of ml.

Reviewer 2 Report

This paper is interesting and well-organized. This paper has some original findings and may give some insight and suggestions to the research community dealing with this subject. However, I have some doubt about the generality of the conclusion of this paper. Followings should be considered:

  1. The plasma exposure time is set at 12 min throughout the experiments. Comparing to the conventional treatment time, it seems a little long, which may give rise to somewhat over dose, especially for direct treatment. This may result in drastic cell death for normal cells.
  2. Why do you choose epithelial cells and human primary skin fibroblast cells as normal cells? Are these cells from the same tissue?
  3. The cell membranes of normal cells and tumor cells are different. It is widely known that tumor cells have more unstable and more vulnerable to external stimuli, which is quite contrary to your discussion. Only a few reports claim to have found nonselective apoptosis-inducing effects of CAP or PAM. Why is normal cell more vulnerable to direct treatment? The authors should provide further comment or references on this issue.
  4. How did you set the incubation time of PAP as 1 h? Do you have some ground for this choice? If the incubation time is prolonged, does the cell viability drop?
  5. You used PBS as the plasma stimulated liquid. Although PBS has been considered as plasma stimulated liquid quite often, plasma activated medium such as DMEM has also been utilized. Is your result generally applicable to other PAM?
  6. In Fig. 1 and Fig. 2, cell viability at 0.5 slm and 1.0 slm differs. Is this due to the difference of nitrite amount?
  7. In Fig. 10, although the magnification is specified in caption, the scale bar is desired in the figure.
  8. As for the recent publication relevant to the topic, the following could be included as reference paper."Jo et al., Anticancer Effects of Plasma-Activated Medium Produced by a Microwave-Excited Atmospheric Pressure Argon Plasma Jet, Oxidative Medicine and Cellular Longevity, 4205640 (2020)"

Reviewer 3 Report

The authors attempted to dissect the short and long-lived species chemistry relevant for plasma cancer treatment. A number of aspects should be addressed before the manuscript can be reconsidered for publication.

Major comments

- At the beginning, two malignant SCC cell lines are compared with one another in the indirect plasma regime (Cal27, more resistant vs. FaDu, more sensitive). Later, to compare the direct versus indirect plasma treatment, the more resistant malignant cell line CAL27 was compared with a non-malignant cell line (RPE-hTERT), why? The conclusion that non-malignant cell lines are generally more sensitive to plasma treatment cannot be made in a comparison with the more resistant CAL27. The discussion also does not go into possible explanations for the different sensitivity of the cell lines in relation to e.g., redox balance, GSH status, etc.

- The wording of the treatment groups in the manuscript is not stringent and should be better explained in the figures. Schemes to delimit the phases of plasma treatment differentiating them from one another, as well as the resulting treatment regimes, would be helpful.

- I discourage the author’s statement to not add the additional data on RONS formation when modulating feed gas flux etc. as these are planned to be included in a second manuscript. Instead, I feel these data should be added. Many of the figures are only made of 1-3 subfigures so there is sufficient space compress several existing figures into fewer figures so that the data of the second manuscript can be added without adding more figures in total. This will make the study more complete, and the source’s chemistry can be better understood.

- The authors end the discussion by stating that the study fully proves Bauer's model. That should be put into perspective. Catalase, NOX1, SOD on the cell surface are not dealt with in the discussion regarding sensitivity either. Moreover, what is the evidence that catalase, NOX1, and SOD are expressed by the cell types investigated?

- The perhaps biggest draw back is the lack of viability data in most experiments. The authors used only plate reader assays that capture the overall (metabolic) activity of cells, which clearly does not allow any conclusions on the viability. As second mode of action, also cell cycle arrest might be present. In human cells that on average divide every 24h-36h, it follows that 30-50% of signal loss reported might be due to cell cycle arrest rather than toxicity. Hence, terminal cell death needs to be shown and quantitatively analyzed for the critical experiments of the manuscript.

Minor comments

- example flow cytometry dot plots (forward scatter vs fluorescence) should be shown to demonstrate the increase in lipid peroxidation from the raw data

- The images in Fig. 10 cannot be clearly seen.

- The title is sometimes missing in the figure legends

- Figure 10 is inconclusive. Cell shapes cannot be seen.

Round 2

Reviewer 3 Report

The authors have addressed many of the comments, but more clarification is needed, see comments below regarding the numbered points raised in the first set of comments.

Major:

#1 The extended discussion helped to assess the point of different effects in different cell lines. Please be also reminded to attenuate the statement that non-malignant cells are generally more robust inside the whole text.

#2 The overview of Fig. 14 indeed helps to understand the different treatment regimens. However, we do not understand why this information is given in the last figure and think it should be implemented very early in the manuscript. Also, we strongly encourage the authors to re-consider their wording. It is hard for the reader to understand the different treatment modalities when "plasma-treatment" is different from "direct- and indirect-plasma treatment." To change this, we propose to update the "direct plasma treatment" branch into "immediate-" and "early effects (1h)" (except for "direct" and simple "plasma-treatment") to make this more logical.

#3 I trust the authors' reply that they are splitting the manuscripts only because this present manuscript is mainly on the biology while the second one will be instead on the physico-chemical relationships of the RONS formation. On the point of the number of figures, multi-panel figures are standard in life sciences and stretching the figures to a total of 14 is the author's choice and not necessarily a quantitative "measure". Moreover, some of the figures are very large, e.g., figure 2, with bar graphs of about 20-30x the size of regular text, which looks very imbalanced.

#4 We think the discussion significantly improved in its quality.

#5 For the manuscript's conclusion, it is critical to have another method validating actual cell death. Still, these experiments were not carried out for essential experiments as in Fig. 11 and 12. A simple imaging or flow cytometry experiment with %PI+ cells would be enough. The CellTiter Glo assay measures, according to its statements, the metabolic activity of the cells by determining ATP (Hannah et al., Promega Cell Notes 2 (2001): 11-13). Information about terminal cell death cannot be drawn. The authors also could simply do a dose-response of their treatment for one cell line, analyze metabolic activity and viability in parallel, and show in a supplemental plot that these data correlate. Also, the cell cycle analysis in figure SF5 is not sufficient and lacks any quantification. This information should also not be hidden in the supplementary information, and simple quantification of G2/G0 cells would help reading the flow histograms. Moreover, the wording "viability" has not been changed in the manuscript as requested, where only metabolic activity was assessed.

Round 3

Reviewer 3 Report

The authors suitably addressed all points and the manuscript should be published.